# The Two-Sided Effect of the COVID-19 Pandemic on Online Apparel Renting

**Hyejune Park [1,\*] and Min-Young Lee [2]**

[1] Department of Design and Merchandising, Oklahoma State University, Stillwater, OK 74074, USA
[2] Department of Merchandising, Apparel and Textiles, University of Kentucky, Lexington, KY 40506, USA
* Correspondence: june.park@okstate.edu; Tel.: +1-405-744-5628

**Abstract:** The global outbreak of COVID-19 has been affecting consumer behavior in a significant way. The collaborative consumption businesses, such as online rental services, is one of those sectors that have been immensely disrupted by the pandemic because many 'sharing' services require high touch and human contact. The purpose of this study is to develop and test a comprehensive model of consumers' acceptance of online apparel renting (OAR) that can also take account of the pandemic's effect. To this end, a two-phase, mixed-method study was conducted to identify the specific determinants of OAR (Study 1) and to empirically test the model of OAR acceptance with the pandemic-related moderating variables (Study 2). This study identifies a series of consumer drivers of OAR including affordable access, shopping convenience, try before you buy, never wear same dress twice, and special occasion. It also demonstrates the significant moderating effects of two pandemic-related factors including perceived vulnerability to disease and desire for simplification. The findings provide practical managerial suggestions to OAR retailers and theoretical implications for future studies.

**Keywords:** online apparel renting; collaborative consumption; COVID-19; pandemic; vulnerability; desire for simplification; technology acceptance model

## 1. Introduction

The global outbreak of the novel coronavirus COVID-19 has been affecting consumer behavior in a significant way. The collaborative consumption businesses such as online rental services is one of those sectors that have been immensely disrupted by the pandemic because many 'sharing' services require high touch and human contact [1–3]. Industry experts claim that the apparel rental service is uniquely impacted by the pandemic. On the one hand, it would experience a reduced demand since consumers now have become more cautious about sharing items with other strangers due to their hygiene and safety concerns [4,5]. On the other hand, the pandemic might bring more values around mindful consumption, and online apparel renting (OAR) could be a more attractive option to those who desire to simplify their consumption by allowing them not to possess too many clothes [6,7]. Yet, it remains unknown whether this two-sided effect of the pandemic is actually at play in the consumer's mind.

Therefore, the current study started with this question: Whether and how the COVID-19 pandemic has affected the OAR business? To answer this research question, the present study first developed the research model that explains the consumer's OAR behavior and then incorporated the pandemic-related factors into the model to test their moderating effects on the relationship between consumer attitude and intention toward using an OAR service.

Consumers have increasingly embraced the idea of online renting for clothing since the concept of OAR service was pioneered by an online apparel renting platform, Rent the Runway, in 2009 [8]. Despite some early concerns about logistical barriers and the need for

consumer behavior shift [9], the global market for OAR has been continuously growing and forecasted to surpass $7 billion by 2025 [10]. In the United States, the value of the OAR market was approximately $1 billion in 2018 and is predicted to reach $3.3 billion by 2025 [11]. The success of OAR has driven established apparel retail brands, such as Banana Republic, Ralph Lauren, and American Eagle, to enter the marketplace to supplement their retail revenue by offering potential customers a way to try their products [12].

While the growth of the apparel rental retail industry is evident, limited research has examined OAR behaviors especially in relation to the pandemic. Nevertheless, prior research provides some useful insights into the drivers of OAR. For instance, researchers have found that OAR is driven by such factors as functional (e.g., getting items more easily and quickly, finding more desirable assortment), economic (e.g., saving money), emotional (e.g., feeling enjoyable), and environmental (e.g., reducing waste through rental) [13–17]. However, with an exception of Park and Armstrong (2019) [17], previous research tends to take an inductive approach to limit predictions only to the intention level rather than the actual behavior level. Further, little research has examined how the pandemic has affected the consumer's perceptions and attitudes towards OAR.

This research attempts to fill this gap in current literature by conducting two studies. In Study 1, we conducted interviews with apparel renters to uncover the specific determinants of OAR. Using the grounded theory approach, the interview data were collected and analyzed to seek theoretical saturation on core concepts (i.e., motivations for OAR) that emerged from the data [18]. In Study 2, the findings in Study 1 were incorporated into the model of consumers' acceptance of OAR based on the Technology Acceptance Model (TAM) [19]. Then, two pandemic-related factors including vulnerability to disease and desire for simplification were added to the core attitudinal model as the critical moderators that may influence consumers' intention to adopt an OAR service. This model was tested using a national consumer sample to increase the validity of research findings. To sum up, this study aims to develop and test a comprehensive model of consumers' acceptance of OAR that can also take account of the pandemic effect.

## 2. Literature Review

### 2.1. Technology Acceptance Model

The Technology Acceptance Model (TAM) developed by Davis (1989) [19] is considered as a reliable and robust model to understand the user acceptance of technologies by predicting usage intention and behavior. Drawing from the theory of reasoned action [20], TAM focuses on two key beliefs, ease of use and usefulness, as the drivers of users' attitudes, intentions, and adoption behavior [21]. The TAM has been applied in consumer research to understand the user's acceptance of technologies in a variety of settings. For example, Ashraf et al. (2014) [22] and Tong (2010) [23] applied TAM to understand the adoption of e-commerce across different cultures. Vijayasarathy (2004) [24] introduced an extended TAM model to predict consumer intentions to use online shopping by adding the variables of compatibility, privacy, security, normative beliefs, and self-efficacy to the existing TAM model. The predictive power of the TAM enables researchers to apply it to various settings and to understand different purchase behaviors particularly in an e-commerce context [25,26].

The TAM theorizes that two key beliefs about new technology including perceived usefulness and perceived ease of use determine a consumer's attitude and/or intention to adopt technology [19]. Perceived ease of use refers to the extent to which a consumer believes that using a new technology would not require any extra effort. Perceived usefulness is defined as a consumer's belief that using a specific technology will increase his or her job performance [27]. The TAM further suggests that perceived ease of use is instrumental in explaining the variance in perceived usefulness [27]. Consistent with the attitudinal research suggesting that attitudes have a positive effect on intentions [28,29], the TAM claims that the more positive attitude consumers have toward using a certain technology, the more likely they intend to use such technology. Prior research supports this contention [24,30,31]. Therefore, when consumers find a new technology easy to use

and useful, they are more likely to have a positive attitude toward using it, which further predicts their adoption of the technology.

### 2.2. Determinants of Online Apparel Renting (OAR)

Prior research has provided empirical evidence on the drivers of OAR. Baek and Oh (2021) [13] used the theory of consumption values [32] in examining the drivers of attitude toward fashion renting and found the functional (e.g., getting the apparel more quickly and easily), economic (e.g., saving money through OAR) and emotional values (e.g., feeling enjoyment) significantly predict attitude toward online fashion renting. Yet, they did not find the environmental value (e.g., reducing pollution through OAR) or social value (e.g., feeling accepted) as a significant predictor. On the other hand, other studies found that ecological benefits did predict consumer attitude toward fashion renting [14,15]. For instance, Lee and Chow (2020) [14] found that the two attributes of fashion renting including ecological benefits (e.g., reducing pollution through OAR) and functional benefits (e.g., getting the apparel more easily and cheaply) were significantly related to the consumer's attitude toward online fashion renting, which in turn predicted intention to adopt fashion renting. In another study [16], a consumer's frugal shopping behavior was explored as a determinant of attitude toward fashion renting, and it was found that a consumer's general tendency to save money and resources predicts attitude toward fashion renting and perceived enjoyment of renting, which in turn predict intention toward fashion renting.

While these studies mainly deducted the drivers of OAR based on theories, Park and Armstrong (2019) [17] used a more inductive approach to uncover the motives of OAR. Through consumer interviews with 14 OAR users, they found that consumers adopted OAR because they perceived that OAR saved their money and time for fashion shopping, was easy to use, and helped them find desirable product assortment through services such as styling. Additionally, they found that consumers adopted OAR because they wanted to try trendy items or special-occasion clothes without having to purchase, switch out the clothes with new items, and try items without having to commit to buying. The current study extends Park and Armstrong's (2019) [17] qualitative work in two important ways. First, this study incorporates the OAR determinants that emerged in a qualitative analysis in the model of OAR adoption with a series of outcome variables to test specific hypotheses developed based on TAM. Second, this study additionally examines the moderating effects of pandemic-related factors on the link between attitude toward OAR and the intention to adopt OAR.

To sum up, previous studies consistently suggest that perceived economic benefits of OAR and other functional benefits, such as saving shopping time and finding items consumers want more efficiently, are the strong drivers of OAR. Yet, several other factors including environmental value or social value (e.g., social norm or feeling accepted) have been found to be inconsistent predictors in prior research. When it comes to research methodology and approach used, little research has utilized a mixed-method approach to overcome the limitation of deductive approaches (i.e., limiting predictions only to the intention level) and qualitative approaches (i.e., data collected are not statistically representative).

### 2.3. The Pandemic's Effects on Online Apparel Renting

2.3.1. Perceived Vulnerability to Diseases

Consumers might be less likely to adopt OAR if they are concerned about the possibility of virus exposure from the goods that have been used by other consumers [4,5]. While there is not enough scientific research to indicate how long the coronavirus can survive on soft surfaces, consumers' concerns about the possibility of virus exposure from the goods that have been used by others may become a psychological barrier to using an OAR service [33]. Indeed, researchers have found that the consumer's concern about contamination of rented goods negatively affects their attitude toward OAR [13,34,35]. Thus, the relationship between attitude and intention toward OAR may be attenuated with greater perceived vulnerability to diseases.

### 2.3.2. Desire for Simplification

Pandemic alerts limited resources both in the finance and in the ecological environment [6,7]. As consumers realize the severe consequences of materialism and overconsumption during the pandemic, OAR involving only the utility of a material good can play in the current shift toward more mindful consumption. That is, those who desire to simplify their consumption may find OAR an alternative to buying and possessing clothes [6,17]. Thus, consumers with greater desire to simplify their consumption through renting may use an OAR service with less reliance on their existing attitudes about this service. In other words, their preexisting attitude toward OAR, whether it is positive or negative, would be less important to them since they believe that renting clothing could help them be more mindful in their consumption.

## 3. Study 1

The purpose of Study 1 is to uncover the specific determinants of online apparel renting behavior. As previously discussed, due to the lack of inductive research on OAR motivations, the consumer interviews were conducted with actual OAR users utilizing a phenomenological interviewing approach [36]. Phenomenology is a qualitative research approach that attempts to describe the meaning of the experience from the perspective of those who had that experience [37]. Hence, the purpose of the interview was to understand the meaning of OAR to those who had an actual experience of OAR.

### 3.1. Methods

#### 3.1.1. Procedure

Given that the majority of current OAR platforms cater to female consumers, this study focused on female consumers. Participants were recruited by sending an email to a random sample of 5000 female faculty, staff, and students enrolled in a U.S. southwestern university. A $15 cash incentive was advertised. Phone interviews were conducted until a saturation in user experience was reached, which included 15 participants. Each interview required 30–60 min. Each interview was audio-recorded and transcribed verbatim for data analysis.

#### 3.1.2. Participants

Ten out of 15 participants were aged 18–24, four participants were 25–30, and one participant was aged 31–40. Although they were largely young consumers, this characteristic was deemed appropriate given that clothing rental users tend to be young [38]. The majority (86.6%) of participants were Caucasian. The largest number of participants ($n = 9$) reported that they used OAR platforms once or twice a year, followed by once or twice a month ($n = 4$), and once or twice per week ($n = 2$). The OAR platforms reported included Rent the Runway, LeTote, Gwynnie Bee, and Bag Borrow or Steal.

#### 3.1.3. Analysis

Interview scripts were analyzed using the grounded theory approach that is useful when generating hypotheses from qualitative data [39]. The interview data were first analyzed with open coding to generate a broad range of themes related to OAR. Using the constant comparative method [40], the initial list of themes and categories that emerged from the data were regrouped and further analyzed to focus on the central topic, that is, consumer motivations for OAR. The themes were refined such that they are linked to a theoretical mode as the determinants of consumers' OAR acceptance.

### 3.2. Results

An iterative analysis of core themes revealed five determinants of OAR, namely, affordable access, shopping convenience, try before you buy, never wear same dress twice, and special occasion.

### 3.2.1. Affordable Access

The most dominant driver of OAR found in our data is the consumer's motivation to save money by renting name-brand or designer clothes at an affordable price ("I can rent really nice things that I might not necessarily be able to buy myself"). The majority of participants stated the benefit of 'affordable access' as the first and most important reason for their engagement in OAR. The economic benefit of renting or other 'sharing' mode (e.g., resale) for clothing is also evident in current literature [13,17]. Further, it was found that financial constraints consumers had often drove them to choose renting over buying for the clothes they wanted. In other words, for some consumers, OAR was the alternative to buying, which enabled them to try the clothes they could not afford, as demonstrated in the following comment: " . . . there's been a lot of things I wished I could keep, but even at a discount, it's still, to me, out of my price range". Thus, OAR seemed a remedy to some consumers for the desire for expensive clothes they cannot afford ("When I see the rented item, I always imagine myself or thinking that, "Will I be able to afford this?" But then I feel happy that I'm at least able to use those").

### 3.2.2. Shopping Convenience

Another frequently-cited driver was the convenience related to saving shopping time and effort. Participants indicated that OAR can eliminate shopping hassles including store visit and product return they used to have in traditional apparel shopping: "It helped me not waste my time because instead of spending my time looking on stores, I would know that I would have a box coming. I would say that it definitely saved time which was a benefit of it". Some participants indicated that convenience provided by OAR significantly changed their apparel shopping such that they have rarely engaged in traditional shopping for clothes since trying an OAR service: " . . . I haven't been actually to a store in a long time, probably a year . . . I haven't really bought anything online from any other place besides Gwynnie Bee"). This driver was particularly prominent among those who need convenience due to the lack of time, such as a working mom (" . . . it's so much more convenient for someone on the go and a single mom like myself whose work is very elaborate with travel. It's all I can do really").

### 3.2.3. Try before You Buy

Some consumers see OAR as a way to shop clothes with an advantage of not committing to buying, as one participant commented that "With renting, you can just try a whole variety of different things before you make any kind of decision. It's not like a decision you have to make immediately". This driver indicates that the consumer motivation for OAR is not only to rent the item for a limited time period and then return it but also to try the item with no further obligation before making a purchase. It was also noted that this driver was particularly dominant among those who frequently have sizing or fit issues, such as those who are plus or petite sizes, as OAR can ease the process of trying different sizes and returning the items, which is otherwise a burdensome process when shopping online: "I am a petite woman and not all clothes fit me as the description says. This one I can try and exchange at any number of times. It's hassle-free".

### 3.2.4. Never Wear Same Dress Twice

One of the drivers that is unique to OAR is the consumer's desire to switch out the clothes with a different, new item without repeating the same dress. To those who do not like to wear the same dress twice, OAR is certainly an appealing option: "I just don't like wearing the same dress twice . . . [because] everybody has seen it so I have to get something new". OAR enables these consumers to alternate their outfit without having to purchase clothes too frequently and to try something new at the fraction of the cost. It was also found that, for some consumers, the very idea of trying new clothes continuously drove them to try an OAR service as it was a new, innovative way to consume clothes. One participant commented: "My key reason was that I just found it appealing—being able to change your

wardrobe every week, every two weeks. You can swap out the items with new things. It just sounded really cool. I didn't actually hear anything like that before. I was like, "I definitely want to try it out". It paid off really good way".

### 3.2.5. Special Occasion

The last determinant that emerged in the data is the consumer's motivation not to buy the clothes that will be worn only once for a special occasion (e.g., an evening dress for a fancy event). This finding is not surprising as traditional apparel rental services for special occasions such as tuxedo rental services had existed even before the OAR businesses began [41]. It makes economic sense for consumers to rent the item rather than buying and keeping it if they would not wear it again. Several participants actually indicated that buying an item for special occasions is the only time that they would rent ("The only time I'd rent something is if it's something that I don't own something similar to . . . The fancier the event, the more likely I'll use it"). Thus, this driver was evident for most OAR users.

## 4. Study 2

The purpose of Study 2 is to develop and test the research model that can predict the consumer's acceptance of OAR while taking account of the COVID-19 pandemic effects. To propose the model of OAR acceptance, first, the core attitudinal model was proposed based on the TAM [17]. As previously discussed, the TAM suggests that perceived ease of use and usefulness of an OAR service will have a positive effect on attitude and intention toward using an OAR service. Thus, the following four hypotheses were formulated:

**H1.** *The perceived ease of using an OAR service will have a positive effect on perceived usefulness of an OAR service.*

**H2.** *The perceived usefulness of an OAR service will have a positive effect on attitude toward using an OAR service.*

**H3.** *The perceived ease of using an OAR service will have a positive effect on attitude toward using an OAR service.*

**H4.** *Attitude toward using an OAR service will have a positive effect on intention to use an OAR service.*

Next, the five OAR determinants that emerged from Study 1 were added to the core model as the direct antecedents of perceived usefulness of OAR. That is, when consumers perceive the following specific benefits of OAR, they will consider OAR as useful. The corresponding hypotheses are as follows:

**H5.** *Affordable access will have a positive effect on perceived usefulness.*

**H6.** *Shopping convenience will have a positive effect on perceived usefulness.*

**H7.** *'Try before you buy' will have a positive effect on perceived usefulness.*

**H8.** *'Never wear same dress twice' will have a positive effect on perceived usefulness.*

**H9.** *Special occasion will have a positive effect on perceived usefulness.*

Further, based on the literature review, the research model was extended with the two pandemic-related factors including vulnerability to disease and desire for simplification. As previously discussed, consumers may be reluctant to engage in OAR if they perceive themselves to be vulnerable to infectious diseases from rented items. Thus, the relationship between attitude and intention would be attenuated with greater perceived vulnerability to diseases. In addition, for those who believe that OAR could help them be more mindful in their consumption, their attitudes toward an OAR service as a determinant of adoption intention will be less important because they may be willing to try an OAR service anyway. Hence, the relationship between attitude and intention will be weakened for these consumers. The following two hypotheses concerning the moderating effects of pandemic-related factors were developed as follows:

**H10.** *With greater perceived vulnerability to disease, the positive relationship between attitude and intention toward using an OAR service will be attenuated.*

**H11.** *With greater desire for simplification, the positive relationship between attitude and intention toward using an OAR service will be attenuated.*

In sum, eleven hypotheses about OAR determinants and the two pandemic-related moderating effects were proposed, as depicted in Figure 1.

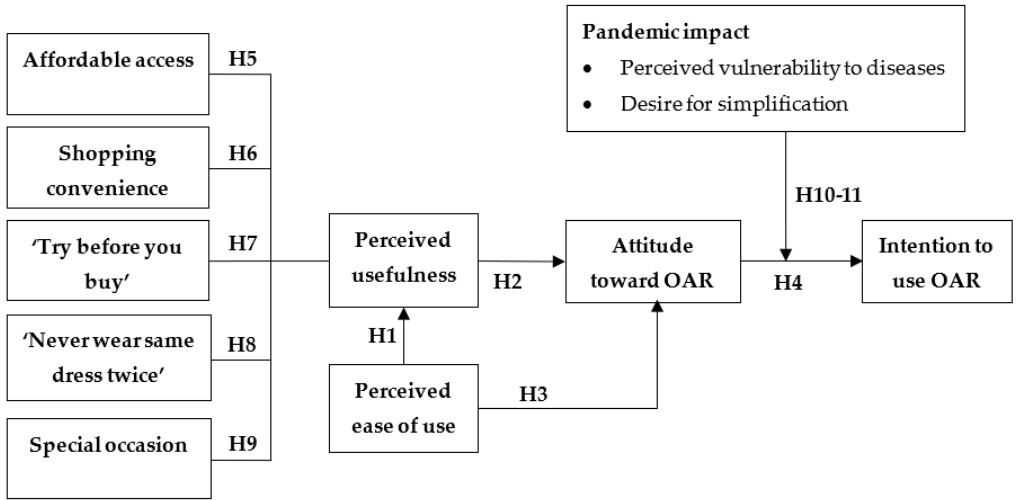

**Figure 1.** Research model.

*4.1. Methods*

4.1.1. Measures

The measurement scales were adapted from the literature and modified to fit the OAR context. For the core attitudinal model, the items for perceived ease of use and perceived usefulness were adapted from Dabholkar and Bagozzi (2002) [42]. The items for attitude toward OAR from Ajzen and Fishbein (1980) [20] were measured using a seven-point semantic differential format (e.g., bad/good, useless/useful). Intention to use OAR was captured using the three items based on Dodds et al. (1991) [43].

For the determinants of OAR, 'affordable access' was measured with three items adapted from Guiot and Roux (2010) [44] (e.g., "OAR service would allow me to access high-end products that I could not ordinarily afford"). 'Shopping convenience' was measured by five items, such as "Using an OAR service would help me save time and effort for apparel shopping" and "Using an OAR service will allow me to shop apparel with less effort" [45]. For the next three constructs ('try before you buy', 'special occasion', 'never wear same dress twice'), the items were developed based on previous literature and findings from the consumer interviews conducted in Study 1. Two researchers who are experts in consumer behavior research worked together to develop, review, and refine the scale items. Each measurement item was carefully reviewed during the data analysis procedure, and we did not find any issue related to the reliability or validity of these constructs. Four items for 'try before you buy' were used: e.g., "Trying products before purchasing via an OAR service will be useful", "An OAR service will prevent me from having to commit to a clothing purchase I am unsure about". 'Special occasion' was measured by three items: e.g., "Using an OAR service will be worthwhile for the clothes that will be worn only once for a special occasion", "It makes a lot more sense to use an OAR service for a fancy dress that I would wear only for a special occasion". The three items for 'never wear same dress twice' included: e.g., "Using an OAR service will be worthwhile because I can change the outfit frequently without buying new clothes", "An OAR service will be useful because it keeps me from wearing the same apparel item twice".

Finally, for the two moderating variables, desire for simplification was measured by four items: e.g., "Renting apparel will help me reduce many clothes that are unnecessary", "It will be worthwhile to rent clothing because I don't need many clothes to live a fulfilling life". The three items for perceived vulnerability to disease was adapted from Duncan et al. (2009) [46]: e.g., "I will not feel comfortable wearing rented clothing because I do not know what the last person who wore it was like". All measurement items except attitude toward OAR were assessed using a 7-point Likert scale (1 = strongly disagree, 7 = strongly agree). The final measurement items were refined based on a content validity test (see the full list of all measurement items and reliability scores in Appendix A).

### 4.1.2. Participants

To increase generalizability of findings, the U.S. national sample was recruited from a market research firm specializing in consumer online surveys. Among 387 female adult consumers, about 83.5% were Caucasian, and the largest number (25.3%) of respondents was aged 51–60, followed by 61–70 (18.6%) and 31–40 (18.3%). Data collection occurred in June, 2020, which was a few months after the pandemic started in the U.S. It must be noted that consumers' perceptions and beliefs about viral diseases might have changed over time. We have addressed this point in the Limitations and Future Research section.

### *4.2. Results*

#### 4.2.1. Assessment of Measurement Model

Confirmatory factor analysis (CFA) was conducted for the measurement model using IBM SPSS AMOS 26.0. One item with low factor loadings (<0.4) [47] and three items with high standardized residual covariance (i.e., absolute values greater than 2.58) [48] were dropped. We confirmed that these items were either reverse-coded items or items that were worded with a slightly different meaning than the other items within the construct. Additionally, we found two pairs of items with high modification indices. Because the correlations between the items were justified by the literature, we allowed those items to freely co-vary. The re-specified model exhibited a good fit: $\chi^2(932) = 2168.847$, $\chi^2/df = 2.327$, RMSEA = 0.059, CFI = 0.952, NFI = 0.919, and TLI = 0.946.

Convergent validity was measured with two criteria. First, CFA loadings for all measurement items were higher than the 0.70 threshold. Second, the average variances extracted (AVE) values for all latent variables exceeded the recommended threshold value of 0.50, ranging from 0.786 to 0.956 [49]. Discriminant validity was also confirmed as the AVE values for all constructs exceeded the shared variance between the constructs [49].

#### 4.2.2. Hypothesis Analyses

The fit indices of the structural model were: $\chi^2(644) = 2043.176$, $\chi^2/df = 3.173$, RMSEA = 0.075, CFI = 0.933, NFI = 0.905, TLI = 0.927. Structural equation modeling (SEM) was conducted to examine causal relationships among constructs. All hypotheses entailing the core attitudinal model were supported except H7 (see Table 1).

**Table 1.** Standardized path coefficients for hypotheses 1–9.

| Hypothesis | Structural Path | Standard Estimate | t-Value |
|---|---|---|---|
| H1 | Ease of use → Usefulness | 0.061 | 2.450 * |
| H2 | Usefulness → Attitude | 0.667 | 17.051 *** |
| H3 | Ease of use → Attitude | 0.404 | 10.654 *** |
| H4 | Attitude → Intention | 0.746 | 17.627 *** |
| H5 | Affordable access → Usefulness | 0.314 | 4.363 *** |
| H6 | Shopping convenience → Usefulness | 0.597 | 10.668 *** |
| H7 | Try before you buy → Usefulness | 0.083 | 0.569 |
| H8 | Never wear same dress twice → Usefulness | 0.268 | 2.422 * |
| H9 | Special occasion → Usefulness | 0.342 | 3.699 *** |

*** $p < 0.001$, * $p < 0.05$.

As for the moderating effects, the multi-group analysis revealed that both moderating effects were significant. First, vulnerability to disease attenuated the effect of attitude toward OAR on adoption intention ($\Delta\chi^2$ = 4.21, $\Delta$df = 1, $p$ = 0.04), supporting H10. Another moderating factor, desire for simplification, also weakened the effect of attitude toward OAR on intention to adopt OAR ($\Delta\chi^2$ = 6.05, $\Delta$df = 1, $p$ = 0.01). Table 2 summarizes the test results of moderating effects.

**Table 2.** Moderating effects of pandemic-related factors.

| Moderating Variable | Structural Path | Hypothesis | Standardized Regression Weight | | $\chi^2$ Difference (df = 1) |
|---|---|---|---|---|---|
| | | | Low Group | High Group | |
| Vulnerability to disease | Attitude → Intention | H10 | 0.761 | 0.702 | 4.212 ** |
| Desire for simplification | Attitude → Intention | H11 | 0.651 | 0.648 | 6.054 ** |

** $p < 0.01$.

## 5. Discussion and Implications

The purpose of this study was to develop and test a comprehensive model of consumers' acceptance of OAR that can also take account of the pandemic effect. To develop a comprehensive understanding of the phenomenon, a mixed-method approach that combines both qualitative interviews (Study 1) and a quantitative survey (Study 2) was used. This study provides important contributions to the current literature by proposing a comprehensive model of OAR acceptance to include the unique motives of OAR and the moderating effects of pandemic-related factors. Findings also provide specific suggestions for OAR service practitioners.

### 5.1. Theoretical Implications

First of all, this study identifies a comprehensive list of OAR determinants through a mixed-method approach. Specific determinants of OAR revealed by actual OAR users demonstrate that OAR is driven by the motives that are clearly distinct from those for other 'sharing' behaviors such as swapping or shopping from resale retailers, corroborating the previous researchers' argument that fashion renting is different from other fashion 'sharing' behaviors [17]. In addition, the current study proposes an integrative model of OAR acceptance based on the TAM [19] and then tests it using a national consumer sample. Data triangulation used in the current study enhanced the validity of research findings and created a more in-depth picture of the OAR phenomenon. Further, the current study advances theory in another important way by incorporating the pandemic-related factors into the model of OAR acceptance. The two moderating variables demonstrate that the pandemic can affect a consumer's fashion renting behavior in a unique way such that different groups of consumers may exhibit different levels of intention to adopt OAR depending on their perceived vulnerability to disease and desire for simplification.

All determinants that emerged in the interview, except 'try before you buy', predicted perceived usefulness of OAR. The significant effect of 'affordable access' and 'shopping convenience' is consistent with previous findings that economic and functional benefits of OAR are the drivers of consumers' attitude toward OAR [13,14,17]. Additionally, the two factors including 'never wear same dress twice' and 'special occasion', which also emerged as the unique drivers of OAR in a previous qualitative study [17], were found to be significant predictors of OAR.

As for the insignificant effect of the 'try before you buy' factor, one possible explanation is that general consumers may not commonly consider an OAR service as a means of product trial. That is, for those who have already used an OAR service (i.e., participants in Study 1), since they already understand the details of an OAR service (for example, there is no fee charged when ordering multiple sizes and returning them), 'try before you buy' can be an important driver of their using OAR. However, for general consumers who lack the understanding of an OAR service (i.e., participants in Study 2), the link between

product trial and usefulness of OAR may be rather weak. Or, given that the study data were collected in the midst of a pandemic, consumers' general unwillingness to try the clothes due to their concern about viral diseases might have influenced this result. As another explanation, it was found in the interview that 'try before you buy' was particularly prominent among those who had a sizing or fit issue. Thus, it is possible that this particular factor is relevant only to a specific segment of consumers rather than to general consumers. Future research may examine this factor further to validate its effect on usefulness of OAR.

Moreover, the inclusion of pandemic-related factors strengthened the core attitudinal model both theoretically and practically. The significant moderating effect of perceived vulnerability to disease indicates that, even with a positive attitude toward OAR, if consumers are concerned about the possibility of virus exposure from the rented items that have been worn by others, that perception can weaken their intention to adopt an OAR service. This result corroborates the previous finding that a consumer's contamination concern about rented goods can serve as a significant moderator of the attitude–intention link [13,35]. Another significant moderator, desire for simplification, suggests that consumers' attitude toward OAR is less important when adopting an OAR service if they believe that OAR can help them be more mindful in their apparel consumption. Therefore, with greater desire for simplification, consumers may be more likely to adopt an OAR service regardless of how they feel about using an OAR service.

### 5.2. Managerial Implications

Findings of this study provide managerial implications for OAR practitioners. First, OAR retailers may use the model of OAR acceptance as a tool to understand a comprehensive picture of consumers' perceptions of an OAR service. The model has been developed based on actual OAR users' experience and verified further by general consumers who may have a varying degree of understanding OAR. Practitioners can also use the specific determinants of OAR in their marketing communication to highlight the benefits of OAR and reinforce the perceived usefulness of an OAR service. For 'affordable access', OAR retailers may emphasize the reasonable cost of trying high-quality luxury brands and offer promotions exclusive to designer labels. For example, Rent the Runway provides their customers with members-only designer deals and exclusive discounts, and this type of strategy could increase the perception of affordable access to luxury items. For 'shopping convenience', OAR retailers may frame OAR as simply a more convenient way of shopping clothes rather than something very different from a traditional way to shop clothes. By deemphasizing the idea of renting and stressing the fact that OAR can eliminate shopping hassle, they may reach a larger group of consumers who have not thought about trying an OAR service before. The 'never wear same dress twice' driver may be particularly appealing to Gen Z and Millennials who like to present themselves with new fashion via social media [50,51]. Indeed, one study found out that many young women feel the pressure to wear a different outfit every time they go out and do not want to wear an outfit again once it has been seen on social media, which may negatively affect the consumer's financial well-being as well as the ecological environment [52]. Thus, offering an OAR service for this group of consumers could be an efficient way to reduce their demand for newly-produced and disposable fashion.

In addition, OAR practitioners should carefully consider the two moderating variables examined in this study. As for perceived vulnerability to disease, while retail businesses including some major OAR retailers report that their businesses have recovered to a pre-pandemic level [53], consumers' perceived vulnerability to disease may still come into play when they consider using an OAR service. Therefore, OAR retailers should provide clear information about how they clean and care for rented items. A detailed description about cleaning processes may alleviate consumers' concern about potential risk of catching an infectious disease from rented goods. This will help mitigate the consumers' concern about the possibility of virus exposure from the rented goods.

With regard to desire for simplification, OAR retailers may stress the role of OAR in reducing unnecessary purchase of clothes and maintaining a minimalist lifestyle. Our findings suggest that those with desire for simplification are more likely to adopt OAR regardless of how they feel about using an OAR service. Therefore, to cater to these consumers, OAR retailers could emphasize such messages as 'fashion freedom' (i.e., enjoying fashion without possessing the items) and 'capsule wardrobe' (i.e., having only essential, staple items in the closet) and explain how OAR can practically help them simplify their fashion consumption. Furthermore, those who seek to simplify consumption through OAR may be more concerned about sustainable consumption of their clothes. Thus, presenting the information about the environmental implications of OAR with scientific evidence could be an effective strategy to engage with this group. It must be noted that the environmental motive did not emerge in the consumer interviews as a driver of OAR, which contradicted previous studies that found a significant effect of environmental motive on OAR behavior [14,15]. This result indicates that consumers engage in OAR not necessarily because they want to reduce pollution or save natural resources. Nevertheless, prior research suggests that OAR can support sustainability by intensifying material utilization and extending the product's natural life [54,55] when it is used responsibly. Therefore, educating consumers about the actual effect of OAR on the ecological environment will positively affect consumers' adoption of an OAR service.

## 6. Limitations and Future Research

This study has several limitations that deserve further research. First, although the current study attempted to triangulate the data by using a mixed-method approach, the findings of this study are limited to U.S. consumers and market. Given that OAR is a global phenomenon that has gained traction in Europe, Asia, as well as America [56], examining the OAR drivers using a wider geographic and cultural area could provide further insights into OAR. Second, in the current study, the two pandemic-related moderating variables were included in Study 2 but not in Study 1 because consumer interviews in Study 1 focused more on uncovering the consumer motives for adopting OAR. To obtain a richer understanding of how the pandemic has actually affected the consumer's preference or perception of OAR, qualitative exploration seems necessary. Third, as previously discussed, data in Study 2 were collected in the midst of the pandemic. Several constructs used in the research model could have been affected by the pandemic circumstance. For example, perceived shopping convenience and usefulness of OAR could have been heightened during the pandemic while the 'try before you buy' benefit of OAR could have been relatively weakened due to the consumer's concern about viral diseases. Researchers could validate the effect of these predictors as the pandemic is now much less severe than it was at the time of data collection. In addition, examining pandemic-related moderators (e.g., perceived vulnerability to diseases) at a different point of time and comparing the results would be another interesting avenue for future research. Fourth, while this study did not directly incorporate sustainability-related factors into the research model, OAR as a business model has interesting implications for sustainability [54]. Future research could examine OAR behaviors using sustainability as a theoretical framework (e.g., triple bottom line of sustainability, four pillars of sustainability) [57] and delve into different consumer segments whose OAR behaviors may be driven by other factors such as social and environmental sustainability. Lastly, future research could refine and extend the model of OAR proposed in the current study by incorporating additional variables such as demographic or other relevant consumer traits. As previously discussed, the insignificant 'try before you buy' factor also deserves further examination.

**Author Contributions:** H.P. designed the study, proposed the methodology, and analyzed the qualitative data. H.P. and M.-Y.L. conducted literature review, analyzed the quantitative data, and wrote the manuscript together. All authors have read and agreed to the published version of the manuscript.

**Funding:** This research received no external funding.

**Institutional Review Board Statement:** The study was approved by the Institutional Review Board of Oklahoma State University (protocol code HE1748, 24 August 2017).

**Informed Consent Statement:** Informed consent was obtained from all subjects involved in the study.

**Data Availability Statement:** The data presented in this study are available on request from the corresponding author.

**Conflicts of Interest:** The authors declare no conflict of interest.

## Appendix A

**Table A1.** Measurement Items and Reliability of Constructs.

| Construct | Items | Composite Reliability |
|---|---|---|
| Affordable access | • Online apparel rental service would allow me to access high-end products that I could not ordinarily afford.<br>• I can wear high-end clothes at a fraction of the cost when using an online apparel rental service.<br>• Online apparel rental service would allow me to access fashion items I really love that are outside my price range. | 0.985 |
| Shopping convenience | • Using an online apparel rental service would help me save time and effort for apparel shopping.<br>• Using an online apparel rental service will allow me to shop apparel with less effort.<br>• Shopping for apparel would be easier when using an online apparel rental service.<br>• Using an online apparel rental service would make my apparel shopping more convenient.<br>• I can spend less time shopping because an online apparel rental service will suggest good choices. | 0.962 |
| Try before you buy | • Trying products before purchasing via an online apparel rental service will be useful.<br>• It will be worthwhile to use an online apparel rental service because I don't have to commit to buy the goods.<br>• Being able to try different sizes and styles of clothing offered by an online apparel rental service will be beneficial.<br>• An online apparel rental service will prevent me from having to commit to a clothing purchase I am unsure about. | 0.991 |
| Never wear same dress twice | • Using an online apparel rental service will be worthwhile because I can change the outfit frequently without buying new clothes.<br>• An online apparel rental service will be useful because it keeps me from wearing the same apparel item twice.<br>• Changing my wardrobe frequently by using an online apparel rental service will enhance my life. | 0.912 |
| Special occasion | • Using an online apparel rental service will be worthwhile for the clothes that will be worn only once for a special occasion.<br>• Using an online apparel rental service will help me keep from buying something I would only wear once.<br>• It makes a lot more sense to use an online apparel rental service for a fancy dress that I would wear only for a special occasion. | 0.945 |

**Table A1.** *Cont.*

| Construct | Items | Composite Reliability |
|---|---|---|
| Perceived usefulness | <ul><li>Using an online apparel rental service will improve my apparel shopping performance.</li><li>Using an online apparel rental service will enhance my apparel shopping effectiveness.</li><li>Using an online apparel rental service would enable me to accomplish my apparel shopping task more quickly.</li><li>Using an online apparel rental service would make it easier to shop for apparel.</li><li>Overall, using an online apparel rental service will be useful.</li></ul> | 0.968 |
| Perceived ease of use | <ul><li>Using an online apparel rental service will . . .</li><li>be complicated/simple.</li><li>be confusing/clear.</li><li>take a lot of effort/a little effort.</li><li>take a long time/a short time.</li><li>require a lot of work/little work.</li></ul> | 0.994 |
| Attitude toward OAR | <ul><li>Using an online apparel rental service will be:</li><li>bad/good.</li><li>foolish/wise.</li><li>unpleasant/pleasant.</li><li>unfavorable/favorable.</li><li>negative/positive.</li><li>useless/useful.</li><li>undesirable/desirable.</li></ul> | 0.989 |
| Intention to use OAR | <ul><li>I would be willing to use an online apparel rental service.</li><li>I would be willing to recommend an online apparel rental service to my friends.</li><li>The likelihood that I would use an online apparel rental service is high.</li></ul> | 0.937 |
| Desire for simplification | <ul><li>Renting apparel will help me reduce many clothes that are unnecessary.</li><li>It will be worthwhile to rent clothing because I don't need many clothes to live a fulfilling life.</li><li>Renting apparel would allow me to avoid buying apparel products just because they are trendy.</li><li>Renting apparel will prevent me from having too many clothes that I don't really need in my life.</li></ul> | 0.936 |
| Perceived vulnerability to disease | <ul><li>I will not feel comfortable wearing rented clothing because I do not know what the last person who wore it was like.</li><li>I am worried about catching an infectious disease from rented clothing even though it has been cleaned by the company.</li><li>I am worried that I might get sick due to some disease transmission from rented clothing.</li><li>I am worried that I might catch some illness from rented clothing that has been worn by others.</li></ul> | 0.964 |

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
