# Peer review of "The Two-Sided Effect of the COVID-19 Pandemic on Online Apparel Renting"

_sustainability, doi:10.3390/su142416771_

Round 1

Reviewer 1 Report

For the authors’ guidance my evaluation and some constructive remarks that would help to improve the paper’s quality are included below:

General Comments

First of all, I would like to congratulate authors for their efforts. There are some interesting and important insights in this article. In more detail, these are the main points that could be improved:

Abstract

The abstract looks fine. It includes analytical and brief information about aim, objective, research findings, and so on.

Introduction

I suggest revising the introductory section of the paper. In the introduction section, the authors ought to reflect the relevant literature review in a more systematic way. Intensively, they may cite more relevant studies and inter-relate them. In the current form, there is a mess in the paper. Please, keep in mind that readers do not have sufficient time. Therefore, the abstract and introduction sections are the most crucial and strategic part that attract the attention of the readers and make them convinced that the paper will not waste their time and it is valuable to continue reading it. Thus, the authors should particularly state what the argument is and how they are testing it methodologically.

Theory-practice nexus is a good tool for a “thesis vs. counter-thesis = synthesis” approach. In this context, the way how the authors support and proof their argument and falsify their counterarguments can be more effective to keep up being focused on the story. So that the central research questions/follow up questions and the “argument vs. counter-arguments interactions” are streamlined in a way that addresses the research inquiries in a systematic and effective manner.

In the light of above-mentioned clarifications, I recommend clarifying the main argument and how the authors deal with it throughout the paper.

Research Questions

The authors did not specify any general/specific research questions. It is better to start with asking questions. The research questions ought to be specific and relevant to the research argument. In case when specific research questions are identified, then the authors can better attract the attention of readers who are able to conceive the theoretical framework of investigation and the authors’ approaches at first glance. I recommend the authors to come up with some research questions, and clarify why the research is interesting and relevant to the field.

Conceptual and Methodological Clarity

The concepts and descriptors used throughout would benefit from more clarity.

Literature review

The key arguments would benefit strongly from further fleshing out.

The authors can enrich the literature review section through adding some up-to-date and relevant scientific materials. In addition, a classification of thoughts, approaches, paradigms etc. will help comprehending the scope and depth of the research.

Methodology

I recommend the authors clarifying the methodology in detail (e.g., research paradigm: positivism/post-positivism/constructivism/critical theory, research design, research tools, and so on), making sure that the planned methods/research tools are fully detailed. They ought to give attention to justifying the chosen methodology and source of data in terms of demonstrating applicability, adjustment, and usefulness in the paper.

 Originality: I have detected a 18% iThenticate Similarity Index Analysis Score. The international standard of tolerance level for peer-reviewed journals is max. 15% (I always recommend the authors to reduce it below 10%). I noticed that there are several passages throughout the paper where phrases and sentences that were taken verbatim from other sources are only referenced with a parenthetical citation when they should also be put in quotation marks. In order for the manuscript to align with the stylistic requirements of the journal, I would ask that the authors ought to fix this issue, appropriately.

Critique

The article largely lacks precision and at times clarity. The argument is not always clear. Moreover, various statements are postulated without clear argumentation and the connection between the points made remains at times obscure. The theoretical and methodological basis for some of the evaluative statements should at time be made clearer.

Concluding Remarks

I appreciate there has been a lot of reading and ground covered. However, the study ought to have a stronger focus, compelling argument and discussion, and an indication of why the paper holds value to the readership of the Sustainability (MDPI). I recommend the authors reconsider the approach adopted here; think about the research questions they wish to examine; make sure the literature review is a lot more cohesive, and make sure the link between the research questions and empirical results is a lot “tighter” than presented herewith.

In light of the above, I think the authors are on the cusp of a very interesting argument and need to probe and deepen it.

Reviewer 2 Report

The article is interesting because of the subject it deals with (Online clothing rental). Also, connecting this topic with COVID-19 makes the article more interesting so far. It is suggested to include the following modifications:

- Given that for various constructs the scales were created or modified from previous scales, it is suggested to present a complete list of the items used in an Appendix.

- It is important to note when (date) the online survey was conducted because consumers evaluated COVID-19 (perception, beliefs) differently over the months. For example, the current perceived risk of contracting COVID-19 from surfaces is much lower than was believed at the beginning of the pandemic.

- It is interesting to see that ‘shopping convenience’ was the main determinant of perceived usefulness. Could the COVID-19 pandemic have affected these results (difficulty making purchases, exchanges and returns in person)? This could also explain the insignificant effect of the "try before you buy" factor. In a context of COVID-19 it is important to clarify this. Once the COVID-19 pandemic is over, could the importance of these factors change?

- Being this a sustainability journal, it would be important to point out in the limitations section that could be important including the four pillars of sustainability: Human, Social, Economic and Environmental as important factors. There may be consumer segments where these factors are more relevant than the factors included in the study.

Reviewer 3 Report

Nicely written paper with appropriate literature and solid methods.

Consider adding an additional literature to the second paragraph in the Determinants of Online Apparel Renting (OAR) section. 

Subtleties between Desire for simplification and Mindful consumption could have some overlap with environmental value, yet this is hard to delineate. This is discussed well at the end of the paper, but could another sentence be added in 2.2/2.3.3 sections to address this.

In the Analysis was any form of inter rater reliability used?

Well thought out Discussion and Implications. Valid limitations that are important to state. The mixed methods approach works well for this study. 

Reviewer 4 Report

Dear author(s),

After reading your interesting paper, it seems far from publication in current version. There are some questions must be addressed, see my comments below for more details:

Specific Comments:

1. The article is very similar to the existing literature, as done by Hyejune P. Cosette H. Joyner Armstrong M. "Is money the biggest driver? Uncovering motives for engaging in online collaborative consumption retail models for apparel”, Journal of Retailing and Consumer Services,2019, 51, 42-50. Accordingly, the contribution of this study is very limited.

2. To avoid the suspicion of plagiarism, in line 360-361,363-368 and 473-475, those paragraphs are necessary to sustainable modification.

3.The structure of the article is weird. Methods in current study are placed in section 3-4, shall be placed on the same section.

 4. In the study, reliability and validity test of results must be conduct.

 5. This article fail to have made a specific contribution to the sustainability issue.

Reference:

Hyejune Park, H.Cosette M. Joyner Armstrong. (2019), "Is money the biggest driver? Uncovering motives for engaging in online collaborative consumption retail models for apparel”, Journal of Retailing and Consumer Services, 51, 42-50.

Round 2

Reviewer 4 Report

Literature Review in the section 2, the writing style or format of citation are required to be extensive edited.  In addtion, the References section must meet the style of the sustainability journal.

Author Response

Literature Review in the section 2, the writing style or format of citation are required to be extensive edited. 

--> We went through the section 2 and edited it. 

In addtion, the References section must meet the style of the sustainability journal.

--> The reference list has been reviewed thoroughly and edited once again to make sure to follow the style of the Sustainability journal. 
